# Riboflavin-Targeted Drug Delivery

**DOI:** 10.3390/cancers12020295

**Published:** 2020-01-27

**Authors:** Milita Darguzyte, Natascha Drude, Twan Lammers, Fabian Kiessling

**Affiliations:** 1Institute for Experimental Molecular Imaging, University Hospital Aachen, Forckenbeckstrasse 55, 52074 Aachen, Germany; mdarguzyte@ukaachen.de (M.D.); Ndrude@ukaachen.de (N.D.); tlammers@ukaachen.de (T.L.); 2Fraunhofer MEVIS, Institute for Medical Image Computing, Forckenbeckstrasse 55, 52074 Aachen, Germany

**Keywords:** riboflavin, vitamin B2, targeted drug delivery, active targeting, theranostics, nanomedicines, molecular imaging, nanoparticle

## Abstract

Active targeting can improve the retention of drugs and drug delivery systems in tumors, thereby enhancing their therapeutic efficacy. In this context, vitamin receptors that are overexpressed in many cancers are promising targets. In the last decade, attention and research were mainly centered on vitamin B9 (folate) targeting; however, the focus is slowly shifting towards vitamin B2 (riboflavin). Interestingly, while the riboflavin carrier protein was discovered in the 1960s, the three riboflavin transporters (RFVT 1-3) were only identified recently. It has been shown that riboflavin transporters and the riboflavin carrier protein are overexpressed in many tumor types, tumor stem cells, and the tumor neovasculature. Furthermore, a clinical study has demonstrated that tumor cells exhibit increased riboflavin metabolism as compared to normal cells. Moreover, riboflavin and its derivatives have been conjugated to ultrasmall iron oxide nanoparticles, polyethylene glycol polymers, dendrimers, and liposomes. These conjugates have shown a high affinity towards tumors in preclinical studies. This review article summarizes knowledge on RFVT expression in healthy and pathological tissues, discusses riboflavin internalization pathways, and provides an overview of RF-targeted diagnostics and therapeutics.

## 1. Introduction

Nanomedicines are nano-sized systems conjugated to anti-cancer drugs. Owing to their size, the nanoparticles accumulate more at the tumor site based on the enhanced permeability and retention (EPR) effect and are expected to show less side effects compared to conventional chemotherapeutics [1]. The EPR effect occurs due to the leaky vasculature and the poor venous and lymphatic drainage of tumors. Nanomedicines can be further functionalized to actively target tumors or their microenvironment. Active targeting can increase the uptake and retention of nanomedicines and, thus, the therapeutic efficacy [2]. The most common targeting moieties are antibodies as well as peptides, aptamers, and small molecules. However, due to their considerably large size, antibodies can significantly alter drug pharmacokinetics and are relatively expensive to produce [3]. Moreover, the coupling of antibodies to drug delivery systems is difficult to control, and their receptor affinities tend to decrease upon conjugation [4,5,6]. Thus, researchers are shifting their focus to small (targeting) molecules, such as vitamins. Among the vitamins, folate-receptors were the most commonly selected cancer targets, particularly for ovarian cancers [7,8,9,10]. However, in recent years, the vitamin B2 (riboflavin (RF)) internalization pathway has also been gaining attention since its carrier protein and three transporters have been identified to be highly overexpressed in several cancers. Therefore, this review article will summarize the current knowledge of the mechanisms of RF internalization and report on studies using this pathway for targeted cancer diagnostics and nanomedicines.

## 2. Riboflavin and Its Transport

RF is a water-soluble molecule that is important for oxidation-reduction reactions [11], protein folding [12], and normal immune function [13,14]. It also has antioxidant and anti-inflammatory properties [15,16]. RF acts as a precursor for flavin mononucleotide (FMN) and flavin adenine dinucleotide (FAD), which are involved in various redox reactions that regulate the metabolism of carbohydrates, amino acids, and lipids (Figure 1) [17]. RF is considered to be relatively nontoxic as an excess of it is excreted via kidneys. Humans do not synthesize RF; thus, they need to get it from their diet. RF deficiency may result in oxidative damage, cell cycle arrest, and cell stress response. It also may impair iron absorption, cause hearing loss and cranial nerve deficits [18,19]. Besides an unbalanced diet, RF deficiency may also occur in inflammatory bowel diseases [20], chronic alcoholism [21], and diabetes mellitus [22].

### 2.1. Riboflavin Carrier Protein

The riboflavin carrier protein (RCP) is not a membrane-spanning carrier but a soluble protein that binds RF; however, the exact role in storing and transporting RF is still unknown. RCP was first identified in oviparous species in the 1960s [23,24]. Chicken RCP (cRCP) has been extensively investigated, as it is easy to isolate and purify in large quantities. cRCP has a high affinity to RF and its co-enzyme forms [23,25,26]. Further investigations using model compounds have indicated that the functionalization of either the isoalloxazine ring or the side-chain of RF results in decreased binding to cRCP [27,28]. During binding, the RF ring is stacked between parallel planes of cRCP [29], while the side chain is oriented inside the cRCP to form hydrogen bonds with it [27]. Furthermore, it was seen that the binding is pH-dependent, confirming the hydrophobic nature of the binding site. Based on these observations, only modification of the C-2 and N-3 positions of the isoalloxazine ring (see Figure 1) should not influence the binding affinity of RF to cRCP [27].

Human RCP (hRCP) shows many similarities to cRCP: molecular size, isoelectric point (pI), and preferential binding to RF over the flavin co-enzymes. hRCP is present during pregnancy and in umbilical cord serum [25]. Suppressing RCP during pregnancy induces abortion in mice and rats, while the well-being of the animals is not affected [30,31]. It is thus assumed that RCP is involved in RF transport to the fetus.

Moreover, overexpression of RCP in patients with malignant disease has been identified. RCP levels in serum were found to be higher in women with breast cancer (6.06 ng/mL) compared to healthy women (0.70 ng/mL) [32]. Another study elucidated that RCP levels correlate with the stage of the disease. Women with early-stage breast cancer had 8.4 ng/mL RCP in serum, 3–4 folds higher than healthy controls (2.8 ng/mL), and patients with advanced tumor stages had even higher levels (20.4 ng/mL) [33]. Similar observations were made in hepatocellular carcinoma, where RCP serum levels were significantly increased (21.75 ng/mL) compared to healthy patients (0.73 ng/mL) [34]. Moreover, RCP overexpression was seen in prostate cancer cells (LnCaP, PC3, and DU-145) in vitro and in tumor tissues in vivo [35]. RCP overexpression in malignant cells makes it a potential biomarker for tumor detection, therapy monitoring, and a promising tool for targeted drug delivery systems.

### 2.2. Riboflavin Transporters

Solute carriers (SLC) transport diverse substrates through membranes such as inorganic ions, amino acids, lipids, and drugs [36,37]. Recently, the SLC52 transporter family has been identified, consisting of riboflavin transporters: RFVT1/SLC52A1, RFVT2/SLC52A2, and RFVT3/SLC52A3 [38,39,40,41,42,43]. RFVT1 and RFVT2 show the most similarity to each other (86.7% amino acid identity), while RFVT3 shows only 42.9% identity with RFVT1 and 44.1% with RFVT2 [43]. RFVT1 is expressed in the placenta, small intestine, plasma membrane, kidney, colon, lungs, uterus, and thymus [38,40]. RFVT2 can be mainly found in the brain, small intestine, and salivary gland [40]. Whereas, RFVT3 is found in the testis, small intestine, and prostate [39,44]. RFVT1, RFVT2, and RFVT3 have Michaelis-Menten constants of 1.38, 0.33, and 0.98 µM, respectively [45]. All three transporters exhibit sodium-independent and temperature-dependent behavior [40,46,47,48,49]; however, only RFVT3 has higher activity in acidic environments; the other transporters do not show pH-dependent behavior [39].

Substances can enter a cell via pinocytosis (Figure 2a), a form of fluid endocytosis, where the cell membrane forms a vesicle encapsulating the fluids and molecules. By this mechanism, small (RF) and large (RF-conjugates) molecules can be transported across the membrane. In addition to pinocytosis, RF is internalized with the help of RCP and RFVTs; however, the exact mechanism remains unclear. It is assumed to be a combination of clathrin-mediated endocytosis (Figure 2b) and carrier-mediated transport (Figure 2c). In the case of clathrin-mediated endocytosis, the substrate binds to receptors (RFVTs) located on the cell surface; and with the help of clathrin, a vesicle containing absorbed substrate is formed. In this way, small and large molecules can be internalized. Moreover, the uptake can be inhibited by receptor saturation. In carrier-mediated transportation, membrane proteins have substrate binding sites that allow specific molecules to pass through the membrane. The substrate size is very important, as it has been shown that upon conjugation to larger molecules, the substrate can lose the ability to pass through the membrane. 

In vitro, high RFVT expression was observed in human epidermoid carcinoma (A431), human renal proximal tube epithelial cells (HK-2), and human umbilical vein endothelial cells (HUVEC) [50]. In vivo, low levels of all three RF transporters have been found in most healthy tissues, whilst tumors seem to overexpress RFVTs: all three transporters were significantly overexpressed in squamous cell carcinoma (SCC), melanoma, and luminal A breast cancer. RFVT3, in particular, was shown to be 187-fold more expressed in SCC compared to healthy tissue [50]. High amounts of RFVT3 were also found in esophageal squamous cell carcinoma, squamous cell carcinoma, and glioma [51,52,53]. Moreover, the SLC42A3 protein level was seen to increase with the stepwise development of esophageal squamous cell carcinoma [54]. RFVTs are thus promising cancer biomarkers and may act as targets for therapeutic drugs or diagnostics, due to their solute carrier function and high overexpression in cancerous tissue.

### 2.3. Riboflavin Internalization

Riboflavin transport processes have been investigated in various cell lines. In all tested cell lines, it was seen that RF uptake is a temperature-dependent and sodium-independent process. The rate of transport decreases with temperature from 37 to 4 °C, whereas the presence or absence of sodium ions in the incubation buffer does not change the uptake kinetics. Additionally, pH-dependent RF internalization was observed in some cases. In human pancreatic cells, RF uptake was increased with decreasing pH from 8 to 6. This cell line has a high expression of RFVT3, which shows better activity at acidic pH [39,55]. Interestingly, in human liver and human retinal pigment epithelial cells, the opposite trend was observed, most likely because these cells function better at a neutral rather than acidic pH. Thus, since tumors are usually characterized by an acidic pH, RFVT3 may be particularly active in tumors and, thus, the receptor to be preferentially targeted.

The kinetic parameters of RF transport vary depending on the cell line (Table 1). It is assumed that this is due to differences in the dominant transport mechanisms, which have not been investigated in detail yet. Among the tested cell lines, the Michaelis-Menten constants (K_m_) of rat brain capillary endothelial cells were almost 50 times higher compared to other cell lines. This means that RF internalization into rat brain capillary cells is an extremely slow process compared to other cell lines. As for the transport rate (V_max_), human intestinal epithelial cells stand out with a relatively high V_max_ value. This is not surprising since humans receive RF mainly from their diet. Taken into account that most other cell lines display K_m_ values below 1 µM, and V_max_ values in the order of fmol/min/mg protein to pmol/min/mg protein, it is assumed that RF transportation is a fast process. This is highly advantageous when using RF as a targeting moiety for diagnostic or therapeutic compounds.

RF internalization can be blocked by competing compounds (Table 2). To test which part of the molecular structure of RF is responsible for blocking uptake, cells were pre-treated with RF or with structural analogs (see Figure 1 for chemical structures) before exposure to RF. In pre-treatment, at least a 20-fold excess of the competing structure was used compared to RF. The pre-treatment with free RF showed that RFVTs can be saturated, reducing RF uptake. The inhibitory effect was also seen when cells were pre-treated with lumiflavin and lumichrome, both of which have an isoalloxazine ring similar to RF. The specificity of the tricyclic isoalloxazine ring to RFVT was further confirmed since bicyclic lumazine did not alter RF uptake. Moreover, pre-treatment with D-Ribose did not saturate RFVTs, indicating that the ribityl side chain of RF is not necessary for internalization. Interestingly, FMN and FAD (co-enzymes of RF) showed varying results depending on the cell line. Though in PBMC cells FMN and FAD did not significantly inhibit RF uptake, a decrease in the uptake was seen. It is likely that the amount of the competing structures was too low to significantly inhibit the uptake. On the other hand, all cells listed in Table 2 have not been tested for their expression of different RFVTs. It could be that different RFVTs have preferred affinity towards RF over FMN or FAD. Thus, blocking with FMN or FAD in the cells expressing these receptors would not be efficient.

Furthermore, different inhibitory compounds have been tested to see if they affect RF internalization (Table 3). Na-K-ATPase inhibitor ouabain showed no inhibitory effect on RF uptake, proving once more that the RF internalization is not sodium-dependent. In contrast, sodium azide and 2,4 –dinitrophenol (DNP) both significantly reduced RF uptake. Sodium azide inhibits oxidative phosphorylation, whereas DNP reduces intracellular adenosine triphosphate (ATP) levels. Hence, internalization is an energy-dependent process. Moreover, the sulfhydryl group modifying agents (p-CMPS and iodoacetate) also inhibited RF internalization. This shows that sulfhydryl groups are important for uptake. Additionally, concentration-dependent inhibition by calmidazolium has been observed, indicating that RF uptake is a Ca^2+^/calmodulin mediated pathway. In general, RF internalization seems to be sodium-independent but energy-dependent, mediated via the Ca^2+^/calmodulin pathway. 

## 3. Riboflavin Targeting

Though RF has started to gain interest as a ligand for active targeting, not much research has been performed thus far. Only a few research groups have synthesized probes functionalized with RF, FAD, or FMN and tested their performance in vitro and in vivo (see Figure 3 for structures). 

### 3.1. Bioconjugates

The first RF conjugate was synthesized in the 1990s [66,67]. RF was covalently linked to bovine serum albumin (BSA) via the ribityl side chain. According to spectrophotometry, five RF molecules were attached per one BSA molecule. BSA-RF and BSA internalization were tested in vitro on human nasopharyngeal carcinoma (KB), human lung adenocarcinoma (SK-LU-1), human ovary adenocarcinoma (SK-OV), and human lung carcinoma (A549) cells [65]. Uptake of the RF conjugate was significantly higher than non-functionalized BSA in all cell lines. Surprisingly, the internalization of the conjugate was not inhibited when cells were pre-treated with an excess of RF or FMN. This is in contrast with analogous studies using other vitamins. For example, in the case of BSA-folate and BSA-biotin, strong inhibition was observed when cells were pre-treated with free vitamins [68,69]. Since other studies using RF targeted systems reported successful competitive binding experiments (see following text), it is likely that either not enough RF was used to saturate the RFVTs or BSA-RF was not taken up via an RF-mediated pathway. The latter could have been due to RF conjugation using the ribityl side chain that has been shown to reduce the binding affinity to RCP [27]. It is likely that BSA-RF could have had higher internalization than BSA due to an increase in hydrophobicity and/or size. Beyond the in vitro study, BSA-RF transport across the distal pulmonary epithelium was tested in vivo [64]. The measurements showed that a higher amount of the conjugate moved from the trachea to blood compared to free BSA. Hence, conjugation to RF seems to increase the transcytosis of BSA through distal pulmonary epithelial cells. Nonetheless, it is unclear if this effect is specifically associated with RF and targeting of its transporters. To address this interesting finding, further experiments are required.

### 3.2. Ultrasmall Superparamagnetic Iron Oxide Nanoparticles

Ultrasmall superparamagnetic iron oxide nanoparticles (USPIO) have been synthesized with FAD or FMN coating for targeted diagnostic imaging [70,71,72,73]. In both cases, USPIO were synthesized via a co-precipitation method and coated with FAD or FMN via phosphate group adsorption. The co-enzymes cannot fully coat the iron particles; thus, guanosine monophosphate (GMP) was also used as a spacer to achieve stable particles. All USPIO had cores of 5 nm according to transmission electron microscopy (TEM) images (Figure 4) and hydrodynamic size of 97 nm for FMN coated [71] and 118 nm for FAD [70]. FMN/FAD coated USPIO (FLUSPIO) had r_2_ relaxivities that were sufficiently high for use as magnetic resonance (MR) contrast agents and close to that of clinically approved agents. 

The particles were further tested in vitro on LnCaP, PC3, and HUVEC. FLUSPIO did not induce toxic effects on cells in concentrations of up to 0.3 µmol Fe/mL according to Trypan blue staining, TUNEL (transferase-mediated deoxyuridine triphosphate nick end tunneling), and MTT (3-(4,5-Dimethylthiazol-2-yl)-2,5-diphenyltetrazolium bromide) assays. MRI experiments showed at least two times higher relaxation rates in cells exposed to FMN USPIO compared to those exposed to USPIO. The competitive binding experiments with FMN decreased the relaxation rates significantly, suggesting an RF-mediated uptake of FMN USPIO. Similar results were observed for USPIO coated with FAD.

In another study, instead of using GMP to produce stable FMN USPIO, ATP, adenosine diphosphate (ADP), and adenosine monophosphate (AMP) were used [73]. The new coatings produced nanoparticle clusters (Figure 4e), which had r_2_ relaxivities higher than that of GMP coated FLUSPIO (Table 4). A cluster consists of a large number of small particles; thus, it is assumed that smaller particles produce higher relaxation rates due to their higher surface area. According to fluorescence measurements, AMP coated FLUSPIO had the highest amount of FMN molecules on its surface, which was likely the reason for the high cellular internalization. Furthermore, ATP, ADP, and AMP did not decrease biocompatibility. In line with this, MR relaxometry using LNCaP, PC3, MCF-7, MLS (human ovarian serous cystadenocarcinoma), A431, and HUVECs showed higher relaxation rate changes when incubated with AMP coated FLUSPIO compared to other FLUSPIO. According to these findings, AMP coating proved to be ideal with respect to good MRI visibility and RF targeting properties. Thus, it is important to find a spacer that ensures stable particles and does not reduce the amount of targeting ligands.

Although FLUSPIO proved promising in vitro, there have been only two studies that investigated their performance in vivo. In both studies, mice bearing LnCaP tumor xenografts were used. The first study compared tumor uptake of FAD USPIO and Resovist^®^ [70]. Instead of R2, R2* relaxation rates were determined in the in vivo experiments due to the higher sensitivity of T2*-weighted MR sequences for iron oxide nanoparticles. MRI scans showed a higher R2* relaxation rate at the tumor site after 1 and 3 h in animals injected with FAD USPIO compared to Resovist® (Figure 5a). It is important to note that FAD USPIO were only compared to Resovist^®^, which has a different core size and different coating that does not exactly match non-targeted USPIO. Hence, the increase in R2* could have multiple underlying causes besides the FAD coating. However, at this time, Resovist^®^ was the only clinically approved diagnostic iron oxide nanoparticle; hence, its use as a control was justified to evaluate FAD USPIO performance against the clinical gold standard. 

Another study investigated the biodistribution and tumor uptake of FMN USPIO. Biodistribution analysis assessed by iron colorimetry showed the highest accumulation at the tumor site followed by the liver and spleen. Competitive binding of FMN USPIO after free FMN administration revealed that 1 and 3 h after injection of FMN USPIO R2*, relaxation rates of tumors were lower when the animals were pre-exposed to free FMN (Figure 5b) [72], suggesting RF-mediated FMN USPIO uptake. This goes in line with in vitro results showing RF-specific uptake of FMN USPIO by PC3 cells and HUVECs [71,72]. Further histological evaluation of the tumor tissue was performed to elucidate which cell types in the tumors predominantly internalize FMN USPIO. Interestingly, FMN USPIO were internalized by tumor cells, endothelial cells, and macrophages. Although the uptake in macrophages and endothelial cells—to a certain degree—can be attributed to unspecific phagocytosis, enhanced uptake can also be the consequence of the upregulation of SLC52 transporters due to the enhanced metabolic activity of these cells [14,50]. In summary, FMN USPIO showed tumor specific accumulation in cancer cells and the tumor stroma, i.e., the tumor associated endothelial cells and macrophages. These results suggest that FMN is a promising diagnostic tag for diagnostic probes to simultaneously target different tumor compartments, including the cancer cells and its stroma.

### 3.3. Dendrimers

The first RFVT targeted drug delivery system was a fifth generation polyamidoamine (PAMAM) dendrimer functionalized with RF [74,75]. The dendrimer (diameter 5.4 nm) was covalently linked to RF via a ribityl side chain and a fluorescent dye or methotrexate (MTX). The fluorescent dendrimer was used to study uptake in KB and HeLa (human cervix adenocarcinoma) cells. FACS quantification showed a dose- and time-dependent uptake of the targeted dendrimers. The XTT assay (2,3-bis-(2-methoxy-4-nitro-5-sulfophenyl)-2H-tetrazolium-5-carboxanilide) using KB cells confirmed the time- and dose-dependent toxicity of PAMAM-RF-MTX. However, the conjugate had lower therapeutic efficacy than the free chemotherapeutic drug (IC50 values after 4 h incubation: 72 nM for the conjugate, 48 nM for MTX). The free RF and RF-PAMAM did not reduce cell viability, indicating that only MTX induced the toxic effects. Moreover, the uptake and, thus, the efficacy of PAMAM-RF-MTX was reduced using PAMAM-RF for competition. Surprisingly, competition with RF did not reduce the therapeutic efficacy of PAMAM-RF-MTX. Similar to the BSA-RF conjugate [67], it could be the case that not enough RF was used for competition. This may be particularly true when considering that, in contrast to competitively blocking with PAMAM-RF, RF may enter the cells via the transporter pathway, which is very fast and efficient so that the carrier-related transport was not significantly affected. 

Furthermore, the RF-targeted dendrimer conjugate (N-10 position) was used to coat gold nanoparticles (AuNP) [76]. The AuNP were prepared by the gold (III) reduction method and had a hydrodynamic size of 30 nm according to dynamic light scattering (DLS), while atomic force microscopy (AFM) showed a 13.5 nm core diameter. The dendrimer conjugates were attached to the AuNP via gold-sulfur chemisorption and had a size of 20.5 nm measured by AFM. The uptake of AuNP-PAMAM-RF was tested in KB cells [76]. According to surface plasmon resonance (SPR) and luminescence of AuNP, the RF-targeted nanocomposite showed higher internalization into cells than the control. Moreover, the interaction of the nanocomposite with RCP was tested using UV-vis spectroscopy. The absorption peak of AuNP-PAMAM-RF decreased depending on the amount of RCP in the solution and shifted by 6 nm to the right (659 nm). In contrast, for non-targeted particles, the absorption peak shifted to the left (622 nm). The distinct different spectral trend suggested that the targeted nanocomposite had specific interactions with RCP. This interaction was further confirmed by AFM measurements of the particle size. The samples showed heterogeneity (two sizes), suggesting that different amounts of dendrimer were absorbed onto the AuNP. Despite this, AFM showed that the particles tended to increase in size when exposed to RCP, thus suggesting interactions between the gold nanocomposites and the RCP. However, as a limitation, the authors reported that AuNP-PAMAM-RF particles need further improvements to yield a homogeneous distribution of RF on the nanocomposites. Nonetheless, this is the only RF-conjugate to date that can be detected using two methods: SPR and luminescence. In conclusion, AuNP-PAMAM-RF is another promising conjugate that could be used for RF-targeted in vitro diagnostics.

Additionally, the same research group also investigated the selectivity of RCP [77,78]. Firstly, it was tested if coupling to different sites of RF changes the binding affinity towards RCP [77]. For this purpose, the PAMAM dendrimer was conjugated to RF via the isoalloxazine ring (N-3 position) or ribityl side chain (N-10 position) (see Figure 1 for isoalloxazine ring numbering). Isothermal titration calorimetry (ITC) and differential scanning calorimetry (DSC) were used to assess the binding affinity of RCP to the dendrimers. The dissociation constant obtained from ITC measurements showed that PAMAM-RF conjugates had a higher affinity to RCP when coupled at the N-3 position than at N-10. Additionally, a higher denaturation temperature was observed for RCP when it was exposed to the dendrimers, confirming increased structural stability. Dendrimer N-3 showed a higher denaturation temperature than N-10, confirming ITC results that the N-3 dendrimer has a better affinity towards RCP. These results support the previous study [27], further confirming that changes at the N-3 position do not alter RF binding affinity to RCP. 

Secondly, the dendrimers with zero, three, or five RF moieties coupled using the N-3 position were used to test if the number of targeting ligands changes, the avidity of the conjugate-RCP complex [78]. The force needed to break the bond between an AFM tip coated with one of the conjugates and the RCP-immobilized surface was measured. Non-targeted PAMAM had a weak attraction to the RCP surface, while in the case of PAMAM-RF with increasing number of RF moieties, higher forces were needed to break the bond. Thus, the study showed that increasing the amount of RF ligands on the dendrimer increases the bond strength of the conjugate-RCP complex. It is possible that at a certain number of targeting ligands the avidity starts to decrease due to steric hindrance. Thus, when designing an RF-targeted system, the number of targeting moieties should be balanced.

### 3.4. Liposomes

The largest structures functionalized with RF so far were liposomes [79]. Here, the RF ribityl chain was substituted by a glycerolipid moiety using a phosphate linker. The resulting RF-phospholipid was then incorporated into liposomes that were prepared by the thin film hydration method. Control and targeted liposomes had hydrodynamic diameters of 115 nm with PDI < 0.1 according to DLS. In vitro experiments showed that A431 cells preferentially internalized targeted over non-targeted liposomes. In detail, the uptake was up to 16 times higher with similar results being obtained for PC3 cells and HUVECs. Furthermore, competitive binding experiments pointed to an RF-mediated uptake. The liposomes did not reduce the viability of A431, PC3, or HUVECs after 72 h of incubation according to MTT assay, proving their biocompatibility. To assess the in vivo performance of RF-targeted liposomes, long circulating liposomes (LCL) were prepared by incorporating polyethylene glycol (PEG) spacers. The hydrodynamic diameter of the control LCL was 137 and 141 nm for targeted ones. A six times higher uptake of targeted LCL was found in PC3 cells compared to control liposomes. In vivo experiments using PC3 tumor xenografted mice indicated that control and targeted LCL had a similar biodistribution and blood half-life. However, histological analyses showed that targeted LCL underwent higher tumor cell uptake than the control ones (Figure 6). This study further confirms that RF-targeting does enhance tumor cell internalization. As liposomes have a high loading capacity and can entrap hydrophobic drugs without altering their structure, RF-targeted liposomes could be an ideal system for targeted drug delivery.

### 3.5. Polymers

In addition to dendrimers, other polymers have been conjugated to RF. One example is poly(N-(2-Hydroxypropyl) methacrylamide) (PHPMA), which was conjugated to folic acid (FA) or RF (via ribityl side chain) and mitomycin C (MMC) to target breast cancer [80]. It was shown that both conjugates (PHPMA-RF and PHPMA-FA) were similarly taken up by SKBR-3 cells, while MCF-7 cells predominantly internalized PHPMA-RF. The MMC-PHPMA-RF conjugate showed significantly higher cellular accumulation than free MMC and MMC-PHPMA in MCF-7 cells. However, when cells were exposed to the lysosomotropic agents monensin and primaquine, only free MMC was taken up. This further proves that RF-conjugates are internalized via an endocytic process. Additionally, the drug efficiency was tested using an MTT assay. The IC50 value of free MMC, MMC-PHPMA-RF, and PHPMA-RF were 0.05 µg/mL, 0.10 µg/mL, and 0.61 µg/mL, respectively. Hence, cytotoxicity derives mainly from the chemotherapeutic drug, and the conjugation to the polymer does not inhibit drug efficacy. Thus, the PHPMA based RF-targeted drug delivery system showed RF specific tumor cell internalization via endocytosis and therapeutic effects associated with MMC. This is the first study to show that RF-targeted drug delivery improves the therapeutic efficacy of anti-cancer drugs in vitro. These promising results encourage further research on RF-targeted drug delivery systems.

Another polymer functionalized with RF was 4-arm PEG (also known as PEGstar) [81]. One of the four arms was labeled with Cy5.5 fluorescent dye, while the other three were conjugated to RF via the N-3 position. In the study, two sizes of PEGstars were used: 10 kDa (7 nm) and 40 kDa (13 nm). The uptake experiments using A431 and PC3 cells showed higher uptake of RF-targeted polymers compared to controls, while competition experiments confirmed RF-meditated internalization. Furthermore, in vivo experiments using mice bearing PC3 or A431 tumor xenografts were performed, showing a longer blood half-life for the bigger particles (40 kDa) due to the lack of renal clearance. Computed tomography/fluorescence molecular tomography (CT/FMT) measurements indicated that 10 kDa RF-PEGstars were retained at the tumor site 3 h post injection, while control polymers dispersed. In the case of 40 kDa PEGstars, both targeted and control polymers were retained in the tumor (Figure 7a). Due to their size, small particles quickly penetrated the tissue but also rapidly redistributed into circulation, leaving only RFVT-bound and internalized particles at the tumor site. In contrast, larger particles passively accumulate at the tumor site and do not rapidly re-enter circulation. While rapid clearance is certainly necessary for a diagnostic probe, intended to display RFVT expression, maximization of overall tumor accumulation may be a necessity for most therapeutics independent of the accumulation mechanism. However, high tumor accumulation does not always mean high tumor cell uptake since the nanomedicines can be accumulated in the interstitial space. In this context, the histological evaluation revealed that targeted PEGstars of both sizes had better internalization compared to controls, clearly indicating the added value of RF-targeting (Figure 7b). Furthermore, 10 kDa RF-PEGstars were mainly internalized by tumor cells, followed by macrophages and endothelial cells, while 40 kDa ones were taken up more by macrophages. It is postulated that the larger polymer was deposited in the tumor stroma for a longer period of time, and thus internalized by macrophages to a higher extent. Nonetheless, high macrophage internalization does not mean that the larger particles would not be a promising drug delivery system; as these cells act as a reservoir of drugs, mediating their slow release towards the cancer cells [82]. In summary, this study revealed a few important facts about targeting with RF and active targeting in general. Firstly, the RF-targeting enhances internalization of the particles into tumor cells but improves tumor retention only for the very small conjugates that do not significantly benefit from EPR. Secondly, the ideal targeted system should be large enough to benefit from EPR but small enough to transit to the tumor cells, which might ideally be given for systems of antibody sizes.

## 4. Conclusion and Future Perspectives

RF seems to play an important role in cancer development and progression, as its carrier protein and transporters are highly overexpressed in several cancer tissues. Thus, RFVTs and RCP have the potential to be used as biomarkers for cancer detection. Furthermore, fast and efficient RF uptake also renders it a promising targeting ligand for cancer diagnostics and therapeutics. In this context, it can be advantageous that both tumor cells and cells of the tumor microenvironment (endothelial cells and macrophages) show enhanced RF internalization. However, the exploration of this interesting pathway is in its infancy, and it is unclear why different cells upregulate different RFVTs and how this influences targeting efficacy. Additionally, the mechanism of RF internalization, depending on the size and composition of the RF-targeted diagnostics and therapeutics, require extensive investigation. Furthermore, the reviewed literature clearly indicates that diagnostic and therapeutic RF-targeted probes require different design considerations. Diagnostic RF-targeted agents should be small molecules with fast exchange kinetics between the tumor compartments and rapid elimination from the body to display the RFVT status. RF-targeted therapeutics, on the other hand, should be larger molecules that strongly accumulate via the EPR but still be small enough to penetrate the tumor tissue and benefit from the enhanced RF-mediated internalization. Thus, if these aspects are taken into consideration, RFVTs may become powerful targets for various theranostic and tumor targeted drug delivery systems.

## Figures and Tables

**Figure 1 cancers-12-00295-f001:**
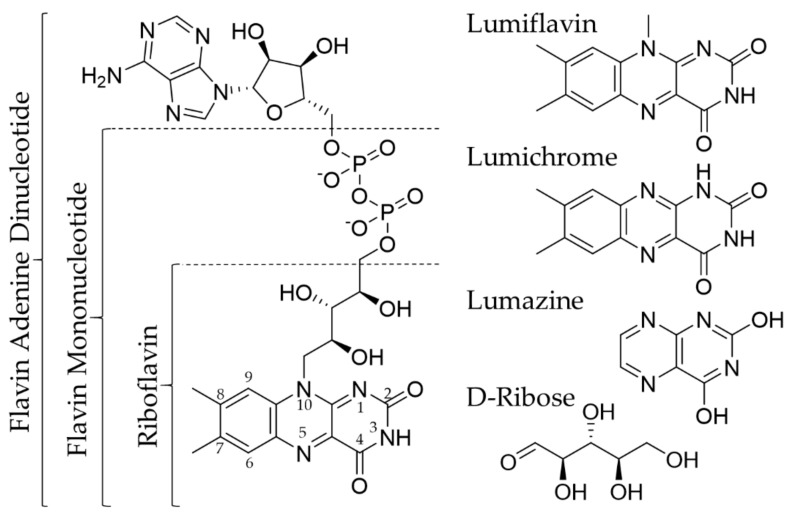
Chemical structures of riboflavin (RF) (with the numbering of isoalloxazine ring), Flavin Mononucleotide, Flavin Adenine Dinucleotide, Lumiflavin, Lumichrome, Lumazine, and D-Ribose.

**Figure 2 cancers-12-00295-f002:**
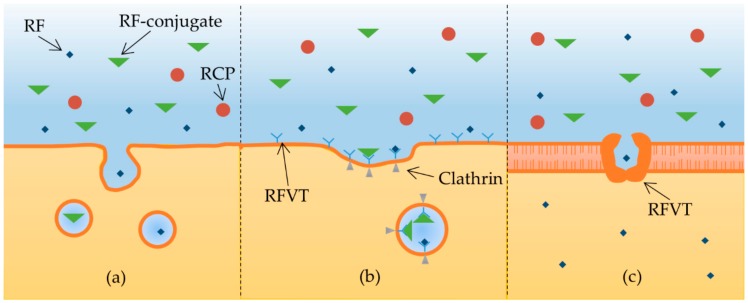
Internalization pathways: (**a**) pinocytosis; (**b**) clathrin-mediated endocytosis; (**c**) carrier-mediated transport. Abbreviations are defined as follows: riboflavin (RF); riboflavin transporter (RFVT); riboflavin carrier protein (RCP).

**Figure 3 cancers-12-00295-f003:**
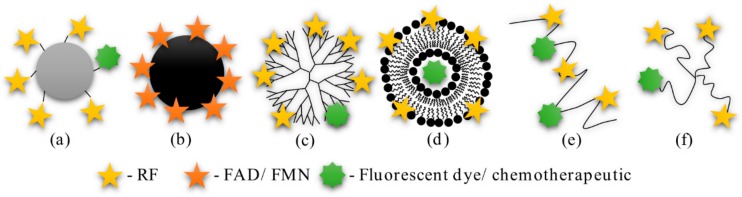
Probes designed to actively target RFVTs: (**a**) Bovine serum albumin-RF conjugate; (**b**) Ultrasmall superparamagnetic iron oxide nanoparticles coated with FMN/FAD; (**c**) Dendrimer-RF conjugate; (**d**) Liposome-RF conjugate; (**e**) Poly(N-(2-Hydroxypropyl)methacrylamide-RF conjugate; (**f**) 4-arm polyethylene glycol-RF conjugate.

**Figure 4 cancers-12-00295-f004:**
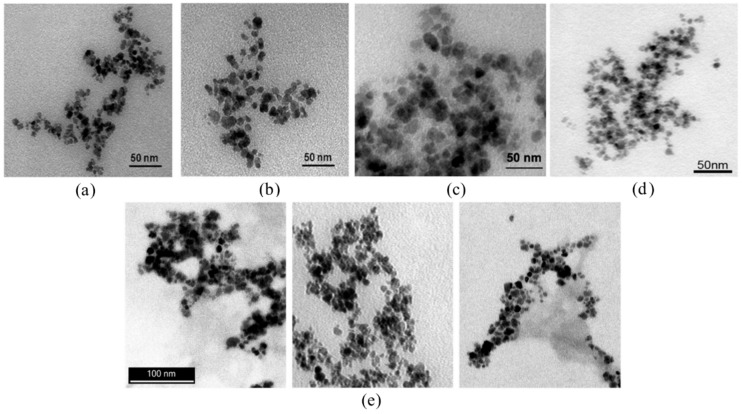
TEM images of different ultrasmall superparamagnetic iron oxide (USPIO) formulations: (**a**) USPIO; (**b**) FMN USPIO; (**c**) FAD USPIO; (**d**) and (**e**) different absorptive coating FMN USPIO: (**d**) GMP-FMN USPIO; (**e**) AMP-FMN USPIO, ADP-FMN USPIO, and ATP-FMN USPIO (left to right); Adapted with permission from [70,71,72].

**Figure 5 cancers-12-00295-f005:**
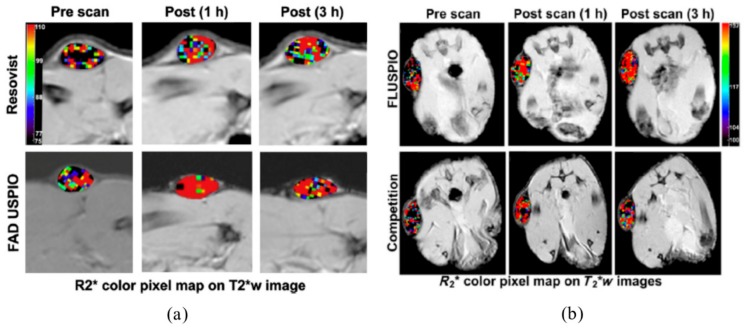
MRI T2 weighted images of subcutaneous right hind limb LnCaP tumors overlaid with color-coded R2* maps: (**a**) FAD USPIO show higher relaxation rates than Resovist^®^; (**b**) the relaxation rates of FMN competition are lower than FMN USPIO (FLUSPIO). Adapted with permission from [70,72].

**Figure 6 cancers-12-00295-f006:**
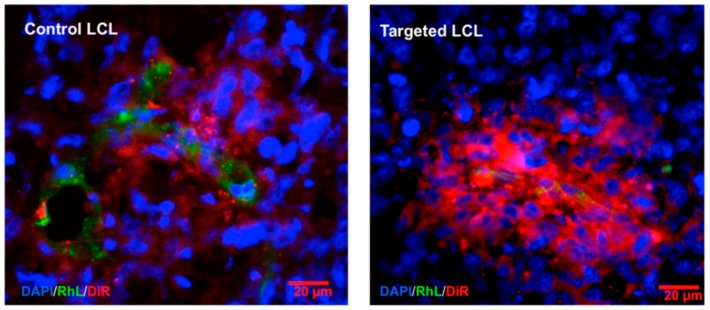
Fluorescence microscopy images of PC3 tumor cryosections (dissected from right hind limb) after 48 h post injection showing higher targeted LCL internalization than control. Liposomes are depicted in red, nuclei in blue, and endothelial cells in green. Adapted with permission from [79]. Copyright© 2020 American Chemical Society.

**Figure 7 cancers-12-00295-f007:**
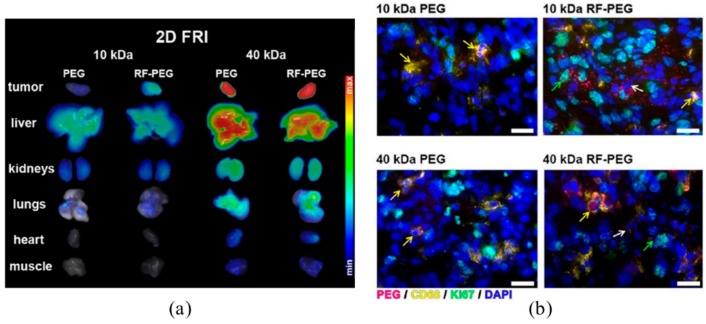
(**a**) Two-dimensional fluorescence reflectance images indicating PEGstar accumulation in mice bearing A431 tumors; (**b**) fluorescence microscopy images of A431 tumors showing the internalization of PEGstars. Polymers are depicted in magenta, macrophages in yellow, proliferating cells in green, and cell nuclei in blue. Adapted with permission from [81]. Copyright© 2020 American Chemical Society.

**Table 1 cancers-12-00295-t001:** Summary of riboflavin uptake kinetics in different cell lines.

Cell Line	K_m_ in µM	V_max_ in pmol/min/mg Protein
Human intestinal epithelial cells (Caco-2) [46]	0.30 ± 0.03	69.97 ± 8.13
Xenopus laevis oocytes [56]	0.41 ± 0.02	0.00005 ± 0.0000007
Human liver cells (Hep G2) [57]	0.41 ± 0.08	1.19 ± 0.08
Human renal proximal tubule epithelial cells (HK-2) [47]	0.67 ± 0.21	3.35 ± 0.29
Human colonic epithelial cells (NCM460) [58]	0.14 ± 0.004	1.10 ± 0.19
Peripheral blood mononuclear cells (PBMC) [59]	0.955 ± 0.344	0.04 ± 0.02
Human trophoblast cells (BeWo) [48]	0.0013 ± 0.00068	0.01 ± 0.001
Human retinoblastoma cells (Y-79) [60]	0.019 ± 0.00037	6.98 ± 0.30
Human retinal pigment epithelial cells (ARPE-19) [61]	0.08 ± 0.014	0.45 ± 0.03
Rabbit corneal epithelial cells (rPCEC) [62]	2.05	3.99
Human embryonic kidney cells (HEK293) [38]	0.0350 ± 0.0041	0.17 ± 0.16
Human breast adenocarcinoma cells (MCF-7) [63]	0.106 ± 0.009	0.52
Human pancreatic cells (β-TC-6) [55]	0.17 ± 0.02	4.45 ± 0.16
Rat brain capillary endothelial cells (BRE4) [64]	19 ± 3	0.24 ± 0.01
Human colorectal carcinoma cells (T84) [65]	0.0532 ± 0.0216	0.36 ± 0.08

Abbreviations are defined as follows: Michaelis-Menten constant (K_m_); maximum uptake rate (V_max_).

**Table 2 cancers-12-00295-t002:** Effects of pre-treatment with competing structures on riboflavin uptake.

Cell Line	RF	FMN	FAD	Lumiflavin	Lumichrome	Lumazine	D-Ribose
Caco-2 [46]	+	nt	nt	+	+	-	-
Oocytes [56]	+	nt	-	+	+	-	-
Hep G2 [57]	nt	nt	nt	+	+	-	nt
HK-2 [47]	+	nt	nt	+	+	-	-
NCM460 [58]	nt	nt	nt	+	+	nt	nt
PBMC [59]	+	-	-	nt	+	nt	-
BeWo [48]	+	+	+	+	+	nt	-
Y-79 [60]	+	nt	nt	+	+	nt	nt
ARPE-19 [61]	+	nt	nt	+	+	-	nt
rPCEC [62]	+	nt	nt	+	+	nt	-
HEK293 [38]	nt	+	+	+	nt	nt	nt
β-TC-6 [55]	+	nt	nt	+	+	nt	nt
BRE4 [64]	+	+	+	+	+	nt	-

Abbreviations and symbols are defined as follows: significant uptake inhibition (+); no uptake inhibition (-); not tested (nt); riboflavin (RF); flavin mononucleotide (FMN); flavin adenine dinucleotide (FAD).

**Table 3 cancers-12-00295-t003:** Effects of different inhibitory compounds on riboflavin uptake.

Cell Line	Ouabain	Sodium Azide	DNP	p-CMPS	Iodoacetate	Calmidazolium
Caco-2 [46]	nt	+	+	nt	nt	nt
Oocytes [56]	-	nt	+	+	nt	nt
Hep G2 [57]	-	+	+	+	+	+
HK-2 [47]	nt	+	+	+	+	+
NCM460 [58]	-	+	+	+	nt	+
PBMC [59]	-	nt	+	nt	-	nt
BeWo [48]	nt	+	nt	nt	nt	+
Y-79 [60]	-	+	+	nt	nt	+
ARPE-19 [61]	nt	nt	+	+	+	+
rPCEC [62]	-	+	+	nt	nt	nt
β-TC-6 [55]	nt	nt	nt	nt	nt	+
BRE4 [64]	+	+	+	nt	nt	+

Abbreviations and symbols are defined as follows: significant uptake inhibition (+); no uptake inhibition (-); not tested (nt); 2,4 –dinitrophenol (DNP); p-chloromercuriphenyl sulfonate (p-CMPS).

**Table 4 cancers-12-00295-t004:** Properties of different USPIO.

USPIO formulation	Hydrodynamic Diameter	MRI r_2_ Relaxivity at 3T
GMP-FMN USPIO [71]	97 nm	203 ± 1 s^−1^mM^−1^
AMP-FMN USPIO [73]	160 nm	228 ± 3 s^−1^mM^−1^
ADP-FMN USPIO [73]	168 nm	233 ± 9 s^−1^mM^−1^
ATP-FMN USPIO [73]	106 nm	259 ± 8 s^−1^mM^−1^
Resovist® [73]	72 nm	233 ± 1 s^−1^mM^−1^

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
