# Peer review of "Riboflavin-Targeted Drug Delivery"

_cancers, 2020, doi:10.3390/cancers12020295_

Round 1
Reviewer 1 Report
This manuscript by Kiessling, F, et al reviews current knowledge in riboflavin-targeted drug delivery with a focus on RF receptors, internalization and potential applications for RF-targeted diagnostics and therapeutics. It provides a comprehensive survey on RF receptors as tumor biomarkers, and RF-conjugated nano systems for therapy and imaging including IONP for MRI imaging and dendrimers for tumor targeting. It is clearly written, and reads well. It will serve as an important article for RF-based drug delivery and imaging. This reviewer recommends its publication in Cancers after consideration of a few minor comments noted below.
Comments:
Cite other reviews articles on RF or its use for drug delivery including: Wong, et al. Riboflavin-Conjugated Multivalent Dendrimer Platform for Cancer-Targeted Drug and Gene Delivery. In Bioactivity of Engineered Nanoparticles, Yan, B.; Zhou, H.; Gardea-Torresdey, J. L., Eds. Springer Singapore: Singapore, 2017; pp 145–171. https://doi.org/10.1007/978-981-10-5864-6_7 The role of RCP seems not sufficiently discussed in section 2.1 Do they have any effect on RF solubilization in water, protecting metabolism and/or RF release on the cell surface prior to intracellular transport (Figure 2)? Reference 32 appears incomplete in its citation for its source.Author Response
Please see the attachment

Reviewer 2 Report
The current review presents an overview of riboflavin (RF) and its role in cancer targeting and detection. This area of research is an area of interest to the cancer research community, and the review highlights interesting nanoparticle/protein constructs with RF and discusses their application in various tumor xenograft systems and cell line models. While the review presents an interesting area, the english should be extensively revised as there are several instances of grammatical issues, missing words, and spelling errors. I have listed some here, but as there were many, some thorough editing is needed. Furthermore, much of the review reads as a list of unrelated facts and experimental findings, with lack of an explanation as to why these facts are important or discussion about implications. For instance what is the significance of showing the KM values in so many different cell lines when it is not further discussed. Unfortunately, I would reject this review in its current form without much more extensive discussion of the status of the field and of the topic. The review does an effective job of listing data from other work, but does not discuss why this data is interesting or important/what the significance is. Without that, the review does not add value over reading the manuscripts. This is the major problem with the manuscript.
Below are some minor recommended changes:
Page 1 line 32 typo should be: Certain sized particles
Line 38: affinities tend to decrease upon conjugation
Line 40-41: However, in recent years, the vitamin B2 (riboflavin (RF)) internalization pathway has also been gaining attention
Line 43 “the current knowledge of mechanisms… “
Check punctuation (esp add in commas throughout)
Page 2: line 49 RF is a precursor for the biosynthesis of FMN and FAD which are…
Line 51: RF is considered to be relatively nontoxic, as excess RF is excreted…
Line 72 page 2: while the well-being of the mother animal is not affected.
It Is thus assumed that RCP is important…
Page 10 line 362 extant should be extent
Non-grammatical:
Page 3 line 81: do you mean RCP overexpression? Isn't RCP expressed even in healthy states?
In this section there is a lot of listing of facts and fold increase of expression but what is the significance?
What is significance of pH dependence on binding?
Table 1: why are some of these values expressed in units of “Per 3 min”? or per 5 min? while others are per minute?
Formatting of the lines on the table should be fixed for better alignment
Table 2: Do all of these cell lines express the same amount of the RFVT? Is that a reason why some of these do not inhibit (i.e. are these data at all the same concentration of the competitors?)
Figure 3: abbreviations should be defined before the figure and what do all the different symbols on these cartoons represent? This is unclear since the symbols are repeated in all the different sub figures of figure 3 so is a yellow star in part a different from the orange star in part b? Why is the green thing in part a in the center of the liposome in part d. These things should be clarified somehow.
Figure 4: authors mention this is xenograft tumor but what are these images of exactly? Where is the xenograft? In general figure legends should be more descriptive throughout the review.
Please be sure to define all definitions page 8 line 278 (if you define AFM you should also define DLS) Define RCP before using it.
Similar comments for liposome section page 9 what kind of xenograft is this? What time point are the sections taken from? While this information is in the primary reference, ideally the review would provide this information so that the reader doesn’t have to look all the information up.
Reviewer 3 Report
In general this is a well prepared review regarding the functionalization with riboflavin for targeting cancer. The reviewer only recommends amending the review to remove needless details, e.g. the color of riboflavin does not contribute towards the manuscript and should be removed.
Reviewer 4 Report
The work entitled "Riboflavin-Targeted Drug Delivery " by Milita Darguzyte et al. is a comprehensive review article summarizing current knowledge of Riboflavine-targeted delivery systems for diagnostic and therapeutics. It is suitable for Cancers audience and publishable after some revision
I pointed out following issues to be addressed by authors:
1) A thoroughly English polishing is necessary by a moderate native speaker revision to enhance importance of the review.
2) Add a table of acronyms at the end of the manuscript for better understanding\following or the article.
3) A SEM\TEM Panel collecting RF conjugates examples would be a important improvement \adding for the paper
4) A moderate update literature is suggested to highlight novel review features
Round 2
Reviewer 2 Report
Thank you for making the changes. It is much improved. I would now recommend this for publication. However there are still just a few minor missing words or grammatical errors if you would like to double check these.